# The Magee 3 Equation Predicts Favorable Pathologic Response to Neoadjuvant Endocrine Therapy in Breast Cancer Patients

**DOI:** 10.3390/cancers16020339

**Published:** 2024-01-13

**Authors:** Carlos Eduardo Paiva, Maria Paola Montesso Zonta, Rafaela Carvalho Granero, Vitor Souza Guimarães, Layla Melo Pimenta, Gustavo Ramos Teixeira, Bianca Sakamoto Ribeiro Paiva

**Affiliations:** 1Department of Clinical Oncology, Barretos Cancer Hospital, Barretos 14784-400, SP, Brazil; vitorsouzaguimaraes@gmail.com; 2Barretos School of Health Sciences Dr. Paulo Prata—FACISB, Barretos 14785-002, SP, Brazil; pazonta@hotmail.com (M.P.M.Z.); rafaelacgranero@hotmail.com (R.C.G.); teixeiragr42@gmail.com (G.R.T.); 3Department of Pathology, Barretos Cancer Hospital, Barretos 14784-400, SP, Brazil; laylamelo1@hotmail.com; 4Learning and Research Institute, Barretos Cancer Hospital, Barretos 14784-400, SP, Brazil; bsrpaiva@gmail.com

**Keywords:** breast cancer, Magee equation, neoadjuvant endocrine therapy, pathologic response, luminal

## Abstract

**Simple Summary:**

Neoadjuvant therapy is a crucial part of breast cancer treatment, but identifying the patients who will benefit the most from it remains challenging. This summary discusses a study of breast cancer treatment, specifically focusing on neoadjuvant endocrine therapy (NET) for postmenopausal, hormone-receptor positive, HER2-negative breast cancer patients. The study aimed to determine whether Magee equations (MEs) could predict how well patients respond to NET. Among the 75 female participants, ME3 emerged as a promising predictor with an AUC of 0.734. This means that ME3 could effectively identify patients likely to respond well to NET. In the analysis, factors such as clinical staging and molecular subtype also showed significant associations with treatment response. Specifically, patients with ME3 values less than 20 were more likely to have a positive response to treatment. The study suggests that using ME3 as a predictive tool could help identify breast cancer patients who are likely to respond positively to NET. This approach offers a cost-effective alternative to the use of Oncotype DX, a commonly used but expensive tool for treatment decision-making. However, larger prospective studies are needed to further validate these findings, which could ultimately improve the selection of the most suitable candidates for NET.

**Abstract:**

Background: Breast cancer (BC) remains a significant health care challenge, and treatment approaches continue to evolve. Among these, neoadjuvant endocrine therapy (NET) has gained prominence, particularly for postmenopausal, hormone-receptor positive, HER2-negative (HR+/HER2−) BC patients. Despite this, a significant gap exists in identifying patients who stand to benefit from NET. The objective of this study was to assess whether Magee equations (MEs) could serve as predictors of response to NET. Methods: This retrospective study included adult patients with invasive BC who underwent NET followed by curative surgery. Assessment of sociodemographic, clinical, and tumor-related variables was conducted. The ME1, ME2, ME3, and ME mean were analyzed to explore their predictive role for NET response. Receiver operating characteristic (ROC) curves were employed, along with the determination of optimal cutoff points. Logistic regression models were utilized to identify the most significant predictors of pathological response. Results: Among the 75 female participants, the mean age was 69.4 years, with the majority being postmenopausal (*n* = 72, 96%) and having an ECOG-PS of 0/1 (*n* = 63, 84%). Most patients were classified as luminal A (*n* = 41, 54.7%). ME3 emerged as a promising predictor, boasting an AUC of 0.734, with sensitivity of 90.62% and specificity of 57.50% when the threshold was ≤ 19.97. In univariate analysis, clinical staging (*p* = 0.002), molecular subtype (*p* = 0.001), and ME3 (continuous = 0.001, original 3-tier: *p* = 0.013, new 2-tier: <0.001) categories exhibited significant associations with pathological response. In the multivariate model, clinical staging and new 2-tier ME3 (<20 vs. ≥20) were included as significant variables. Conclusions: Patients with ME3 < 20 have a higher likelihood of presenting a pathological response, offering a cost-effective alternative tool to Oncotype DX. Larger future studies with a prospective design are awaited to confirm our findings.

## 1. Introduction

Neoadjuvant endocrine therapy (NET) is a valuable clinical strategy in the treatment of postmenopausal, hormone-receptor-positive, HER2-negative (HR+/HER2−) breast cancer (BC) patients, especially those with strong estrogen receptor expression. Three previous phase 2 studies [1,2,3] have compared NET with neoadjuvant chemotherapy for the treatment of patients with HR+/HER2− tumors, and none of the studies demonstrated superiority between the two modalities. In the study by Alba et al. [1], NET was found to be inferior to neoadjuvant chemotherapy only in premenopausal women, and likely inferior in those with a higher proliferative index. However, in clinical practice, NET has been underutilized [4], reserved for older patients or those with significant comorbidities, raising the question of how to selectively identify patients who would benefit most from this therapeutic approach.

The 21-gene expression-based Oncotype DX (ODX) Recurrence Score (RS) categorizes tumors into low, moderate, and high-risk scores [5]. It is a widely accepted and validated tool in treatment decision-making for early luminal breast cancer (BC) patients in the adjuvant setting [5,6]. Its role has been explored in the neoadjuvant context to predict responses to both chemotherapy and endocrine therapy [7,8]. Regarding the latter, previous studies [8,9] have consistently demonstrated that tumors with an RS below 18 are more likely to respond positively to neoadjuvant hormone therapy, whereas those with an RS between 18 and 25 or higher than 25 have a significantly lower likelihood of responding. These findings suggest that the RS can serve as a valuable indicator for guiding treatment decisions, aiding clinicians in selecting patients who are likely to benefit from neoadjuvant endocrine therapy.

Considering the high cost and limited availability of ODX, multivariate prediction models for the RS were developed and referred to as Magee equations (MEs) [10,11]. In this context, MEs can be considered practical alternatives for identifying patients who could be spared from undergoing the Oncotype DX test while maintaining accuracy in predicting treatment response. Previous studies have demonstrated that ME scores are highly correlated with ODX’s RSs, and decision algorithms based on ME scores have been tested in clinical practice [12,13].

However, to date, no study has specifically addressed the predictive role of MEs in the context of NET. The hypothesis of this study is that Magee equations can serve as readily available alternative predictors of response to NET and may prove valuable for selecting ideal candidates for NET. The present study aims to fill this gap through investigating the potential of MEs as a predictive tool to identify strong responders to NET among luminal breast cancer patients.

## 2. Materials and Methods

### 2.1. Study Design and Setting

This is a retrospective cohort study conducted at the Department of Clinical Oncology, Breast and Gynecology Division, at the Barretos Cancer Hospital (BCH) in Barretos, São Paulo, Brazil. At present, BCH is widely regarded as a leading reference in cancer treatment throughout Latin America, offering publicly funded care to more than 6000 patients daily from all across Brazil.

### 2.2. Ethical Aspects

The present study followed the guidelines set forth in resolution 466/12 of the Brazilian National Health Council and obtained approval from the Research Ethics Committee at BCH (CAAE 54022921.0.0000.5437). Recognizing that a significant number of prospective participants reside at considerable distances and visit the hospital for follow-up every six months, the Ethics Committee granted permission to initially obtain verbal consent via telephone. Following this, the informed consent form was electronically sent to patients who willingly chose to participate.

### 2.3. Participants’ Accrual and Data Collection

Patients aged 18 and older of both genders with invasive carcinoma of any histological subtype, showing positive expression of estrogen receptor, and/or progesterone receptor, regardless of HER-2 expression, who received any NET and subsequently underwent potentially curative surgery were included. Cases with distant metastases, previous invasive breast carcinoma, those operated on in another facility, or those lacking complete information regarding estrogen receptor, progesterone receptor, histological grade, or Ki-67 evaluation were excluded.

Potentially eligible patients were identified in the oncology department’s database concerning patients who received initial neoadjuvant endocrine therapy. Subsequently, they were evaluated for eligibility and invited to participate in the study. Clinical information was extracted from patients’ medical charts by two trained resident oncologists. All data were directly entered into REDCap electronic forms [14]. The form used for data extraction is available in the Appendix A.

### 2.4. Analyzed Variables

Sociodemographic information: Age (continuous variable); menopausal status (pre-/peri- versus postmenopausal); gender (female versus male).Performance status: Functional performance measured using the Eastern Cooperative Oncology Group (ECOG) performance status scale, ranging from 0 (best functionality) to 5 (worst functionality) [15].Tumor stage: initial anatomic clinical staging grouped according to the TNM AJCC 8th edition staging system.Treatment-related variables: Type of medication used in neoadjuvant therapy; adjuvant chemotherapy (yes versus no); type of adjuvant chemotherapy; type of surgery (mastectomy versus breast-conserving surgery); type of adjuvant hormone therapy; adjuvant radiotherapy (yes versus no); duration of hormone therapy use (continuous variable, in days).Tumor pathology-related variables: Histological type (no special type (NST) versus lobular versus mucinous versus other types); Nottingham histological grade (ranging from 0 to 9 and classified as grades 1, 2, or 3); estrogen receptor (positive, if at least 1% of tested cells are positive versus negative); progesterone receptor (positive, if at least 1% of tested cells are positive versus negative); HER2 (positive versus negative versus equivocal); Ki-67 (continuous variable using a cut-off of 20%); molecular subtype (luminal A versus luminal B). ESMO criteria were used to define luminal A and B [16]. The H-Score for both ER and PR was calculated by multiplying the intensity of immunohistochemical staining (0 = absent, 1 = weak, 2 = moderate, 3 = strong) with the percentage of staining (ranging from 0 to 100). Thus, the H-Score for both ER and PR ranges from 0 to 300. Pretreatment tumor size was obtained from clinical records as the maximum unidimensional measurement. The preferred order for selecting the imaging modality for measuring tumor size was as follows: magnetic resonance imaging, ultrasound, or physical examination.Magee equations: Three multivariate models (Magee equations (MEs)) and an average score calculation are available. The MEs can be computed using a free online calculator (https://path.upmc.edu/onlineTools/mageeequations.html (accessed on 20 October 2023)). ME1 utilizes data on tumor size, Nottingham score, estrogen receptor (ER), progesterone receptor (PR), HER2, and Ki-67. ME2 employs similar data as ME1 but excludes Ki-67. ME3 relies solely on ER, PR, HER2, and Ki-67 for computation. The MEs were analyzed in three different ways: as a continuous variable, based on three categories from previous studies (original 3-tier; < 18 vs. 18–31 vs. > 31), and using a cutoff defined through the present analysis into two categories (new 2-tier, < or > the best cutoff point determined using the ROC curve).Response assessment: The pathological response was categorized as follows: complete pathological response (yp0), indicating no residual invasive carcinoma in the breast and axillary nodes, or PCR/minimal residual disease (PCR/MRD), which encompasses the occurrence of either yp0 or ypIA. Pathological stage group IA was included as an outcome, taking into account that PEPI score 0 includes tumors up to 2 cm and negative lymph nodes, i.e., ypIA.

### 2.5. Statistical Analysis

Pre-treatment demographic and clinical characteristics were described in absolute numbers and percentage values, as well as measures of central tendency and dispersion. Data normality was assessed using the Shapiro–Wilk test in conjunction with an evaluation of data distribution patterns.

ROC curve analyses were conducted to identify the predictive role of the MEs and potential cut-off points. Sensitivity, specificity, positive predictive values, and negative predictive values were calculated. Additionally, suggested cut-off points as per the Magee decision algorithm were assessed. Both MEs (categorical) and other potential predictor variables were associated with therapeutic response through Chi-square or Fisher’s exact tests. Continuous variables (age, MEs scores, NET duration) were associated with the pathologic response using the non-parametric Mann–Whitney test. Variables with *p*-values < 0.2 were included in backward stepwise logistic regression models to identify the optimal predictive model, with initial clinical staging and age as adjustment variables. 

As a sensitivity analysis, we conducted ROC curve analyses, association analyses, and logistic regression excluding cases where the initial clinical staging group was TNM IA. This analysis was performed to assess whether cases with early-stage tumors were biasing the analysis, given that pathological response included yp0 and ypIA.

A value of *p* < 0.05 was considered statistically significant. For statistical analyses, SPSS software version 21 was used.

## 3. Results

### 3.1. Patient Characteristics

Out of the potential 120 cases that had received NET and underwent a Ki67 assessment in the initial biopsy, 75 could be classified using ME3, while 61 could be classified using other MEs (ME1, ME2, and MEmean), and were included in the study.

Out of the 75 patients included in this analysis, all were female (*n* = 75, 100%), with a median age (p25–p75) of 69.05 (62.02–77.21) years. The majority were postmenopausal (*n* = 72, 96%) and had an ECOG-PS of 0/1 (*n* = 66, 88%). Of the total, 41 (54.7%) and 34 (45.3%) were classified as luminal A and luminal B, respectively. The most frequent initial clinical staging group was IIA (*n* = 28, 37.3%), with 26 (34.7%) of the patients having clinically involved regional lymph nodes; 21 (28%) and 5 (6.7%) classified as cN1 and cN2, respectively. Table 1 provides details of the study participants’ characteristics.

### 3.2. Factors Correlating with Pathological Response

Initially, ROC curve analyses were conducted to identify the best ME as well as the best cutoff points. Considering the occurrence of only three cases of PCR, ROC curve analyses were performed only in relation to the occurrence of PCR/MRD. The AUC values with their respective 95% confidence intervals were 0.752 (0.623 to 0.856), 0.714 (0.581 to 0.824), 0.734 (0.617 to 0.832), and 0.750 (0.620 to 0.854) for ME1, ME2, ME3, and MEm (Figure 1). When compared to each other, none of the models was significantly better than the other (Appendix A). Therefore, considering the larger sample size for ME3 (*n* = 72) compared to the other MEs (*n* = 59) in this study, we chose to proceed with the analysis focusing on ME3. Additionally, we believe that ME3 has greater potential for clinical utility, as it does not require tumor size for the analysis, which is often a bias due to being obtained using different methods (physical examination, ultrasound, mammography, or MRI) that are not necessarily concordant.

Regarding ME3, the Youden index J was 0.4813 with a threshold of ≤ 19.97. This resulted in a sensitivity of 90.62% (95% CI: 75% to 98%) and a specificity of 57.50% (95% CI: 40.9% to 73%). Additionally, the positive likelihood ratio (+LR) was 2.13 (95% CI: 1.46 to 3.11), while the negative likelihood ratio (−LR) was 0.16 (95% CI: 0.054 to 0.49).

Sensitivity analyses were conducted excluding 16 cases with clinical TNM stage IA tumors. The AUC values (95% CI) were 0.792 (0.649 to 0.897), 0.773 (0.627 to 0.882), 0.778 (0.647 to 0.878), and 0.794 (0.651 to 0.898) for ME1 (*n* = 47), ME2 (*n* = 47), ME3 (*n* = 56), and MEm (*n* = 47), respectively (Appendix A). No differences were identified among the curves (Appendix A).

### 3.3. Association between ME3 and Pathologic Response

The occurrence of PCR/MRD was statistically significantly associated with initial clinical staging group (*p* = 0.001), molecular subtype (*p* = 0.002), as well as with ME3 categories (new 2-tier, *p* < 0.001 and original 3-tier, *p* = 0.013), and ME3 as a continuous variable (*p* = 0.001) (Table 2).

Therefore, multivariate logistic regression analyses were conducted. The final model included initial clinical staging and new 2-tier ME3 with a cutoff of 20 (Table 3).

## 4. Discussion

In the present study, we assessed for the first time the potential role of MEs to aid in the selection of better candidates for NET. ME3 appeared to be of greater practical utility as it does not require the inclusion of tumor size, which has the potential for clinical bias due to being measured in clinical practice in different ways that may not necessarily agree (from caliper measurements to resonance measurements). Patients with an ME3 less than 20 showed a 9-fold higher association with pathological response, and in logistic regression analysis, it was more relevant than categorization between luminal A and B using immunohistochemistry.

In the field of oncology, there is a growing emphasis on the pursuit of personalized treatment strategies. As per earlier discoveries from the TAILORx trial [5], it was determined that there is no advantage in using chemotherapy in the case of women aged 50 and above who have HR+ node-negative BC with an RS between 11 and 25. Additionally, findings from the RxPONDER trial [6] revealed that postmenopausal women diagnosed with HR+ node-positive (specifically 1–3 positive nodes) BC and an RS of 25 or lower can confidently forgo chemotherapy.

Since the use of molecular tools has been consistently demonstrated to be valuable in therapeutic decision-making in the adjuvant setting, the quest for novel tools that aid in safely identifying patients who can be spared from neoadjuvant chemotherapy is a current area of active research. A combined analysis of two phase 2 studies [8] with similar methodology demonstrated that chemotherapy can be omitted without detriment in terms of clinical and pathological therapeutic response in the neoadjuvant treatment of HR+/HER2- BCs with RS between 0 and 25. In the TransNEOS study [9], HR+/HER2- postmenopausal, clinically node-negative BC, the RS was validated as a predictor of clinical response to neoadjuvant letrozole. Lower RSs (< 18) were associated with higher clinical response rates and a greater likelihood of breast-conserving surgery following neoadjuvant treatment, suggesting its utility in treatment decision-making. The SAFIA phase 3 trial [17] incorporated the concept of evaluating the RS prior to commencing neoadjuvant treatment. Endocrine therapy with fulvestrant and goserelin, either with or without palbociclib, was administered to patients with BC and an RS < 31. Following 8–9 months of NET, a clinical benefit of 96% was realized. However, performing a 21-gene assay on biopsy samples immediately did not appear feasible. In such scenarios, the utilization of strategies such as ME calculations could prove to be highly valuable.

A previous study using pre-treatment tumor biopsies has demonstrated a positive correlation (Rho = 0.58) between the continuous scores of MEs and ODX-RS [18]. In a sample of low-grade invasive BCs, commonly hormone-sensitive, the overall agreement between ODX-RS and ME risk categories was 68.7% [19]. While not yet assessed in the NET setting, previous studies have confirmed the predictive role of MEs (particularly ME3) in the neoadjuvant chemotherapy scenario. Farrugia et al. [20] identified that patients with ME3 > 31 were 13 times more likely to achieve a PCR compared to those with ME3 scores less than 31. A subsequent multicenter study [21] found that the PCR rate was 0% in ME3 < 18, 0% in ME3 18–25, 14% in ME3 > 25 to < 31, and 40% in ME3 scores of 31 or higher. Tumors with a lower likelihood of responding to chemotherapy are precisely those where we observe a higher likelihood of response to NET, enhancing the practical potential of using ME3 in selecting patients for neoadjuvant treatment.

This study has several limitations. The primary limitation is that the pathological response included not only PCRs but also tumors classified as ypIA, which are small tumors with a better prognosis. Ideally, pathological response could have been assessed according to Preoperative Endocrine Prognostic Index (PEPI) criteria [22]. Unfortunately, in the present retrospective study, the estrogen receptor and Ki-67 immunohistochemical status of surgical specimens was not available. A clinical trial is currently ongoing at our institution (ANNE trial) where ME3 can be compared to the pathological response according to the PEPI score. Furthermore, response assessments can be carried out through physical examination (using calipers), breast ultrasound, and breast MRI. However, such response evaluation may be biased in retrospective studies where the chosen examination and timing of assessment were not planned in advance. Another limitation is the small sample size; however, the use of NET is generally uncommon (for instance, only 3% in the USA [4]), and larger sample studies are typically challenging and often require multicenter collaboration. The retrospective data collection design introduces another potential bias, as the patients’ histopathological reports were not systematically reviewed for immunohistochemical markers.

On the other hand, the main strength of our study lies in its originality, as MEs have not yet been assessed in the NET setting. Our results are hypothesis-generating but require further confirmation in prospective studies with a larger sample size. Additionally, the assessment of pathological response is expected to be conducted using the more widely validated PEPI score.

## 5. Conclusions

In this current study, the application of Magee equations has demonstrated clinical utility in identifying luminal BC patients with a greater probability of achieving a pathological response to NET. Patients with an ME3 score below 20 exhibited a 9-fold higher rate of pathological response compared to those with an ME3 score equal to or greater than 20. Confirmatory prospective studies with larger sample sizes are eagerly anticipated.

## Figures and Tables

**Figure 1 cancers-16-00339-f001:**
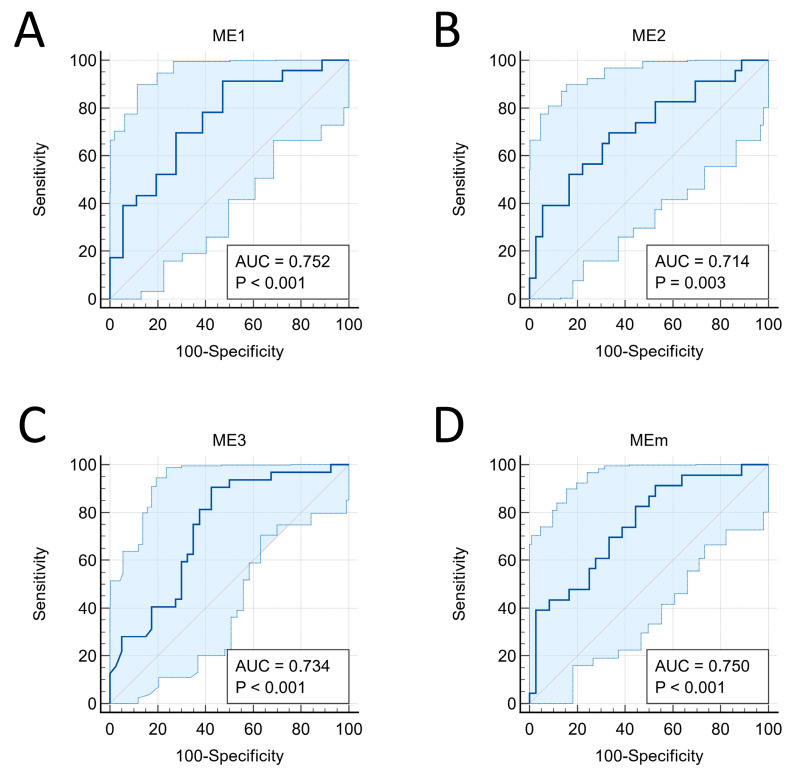
ROC curves with respective confidence intervals and AUC values. (**A**) ME1; (**B**) ME2; (**C**) ME3; (**D**) MEm. The blue ranges represent the 95% confidence bounds for the ROC curves.

**Table 1 cancers-16-00339-t001:** Clinical characteristics of patients included in the study.

Characteristics	N	%
Menopausal status		
Peri/premenopausal	3	4.0
Postmenopausal	72	96.0
Histologic type		
Ductal/No special type	65	86.7
Lobular	3	4.0
Mucinous	2	2.7
Other	5	6.7
Nuclear grade		
I	8	10.7
II	47	62.7
III	18	24.0
Missing	2	2.7
Nottingham grade		
I	23	30.7
II	43	57.3
III	8	10.7
Missing	1	1.3
Estrogen receptor		
Negative	1	1.3
Positive	74	98.7
Progesterone receptor		
Negative	8	10.7
Positive	67	89.3
HER2		
Negative	73	97.3
Positive	2	2.7
ECOG-PS		
0	33	44.0
1	30	40.0
2	6	8.0
3	2	2.7
Missing	4	5.3
Initial TNM clinical stage group		
IA	16	21.3
IIA	28	37.3
IIB	9	12.0
IIIA	6	8.0
IIIB	16	21.3
Molecular subtype		
Luminal A	41	54.7
Luminal B	34	45.3
Endocrine therapy medication		
Anastrozole	67	87.2
Letrozole	6	7.4
Exemestane	6	7.4
Tamoxifen	2	2.5
Surgery type		
Mastectomy	26	34.7
Breast-conserving	41	54.7
Other	6	8.0
Missing	2	2.7
Pathological response (PCR)		
Yes	3	4.0
No	69	92.0
Missing	3	4.0
Pathological response (PCR/MRD)		
Yes	32	42.7
No	40	53.3
Missing	3	4.0
Characteristics	Median	p25-p75
Magee equations		
ME1	18.06	12.89–24.12
ME2	16.35	12.85–22.21
ME3	17.80	12.79–21.51
Mem	18.00	12.59–22.48

Legend: ECOG-PS = Eastern Cooperative Oncology Group Performance Status; PCR/MRD = pathological complete response plus minimal residual disease. ME1 = Magee equation 1; ME2 = Magee equation 2; ME3 = Magee equation 3; MEm = Magee equation mean.

**Table 2 cancers-16-00339-t002:** Association between clinical and pathological variables and the occurrence of pathological response.

Variables	Pathologic Response #		*p*-Value
	Yes	No	
Age (years), median (p25–p75)	65.46 (62.47–77.02)	69.37 (61.22–78.17)	0.493 ³
Menopausal status, *n* (%)			0.581 ¹
Pre or peri	2 (6.25)	1 (2.5)	
Post	30 (93.75)	39 (97.5)	
Histologic type, *n* (%)			0.326 ¹
Ductal	30 (93.75)	32 (80)	
Lobular	0 (0)	3 (7.5)	
Mucinous	1 (3.12)	1 (2.5)	
Other	1 (3.12)	4 (10.0)	
Nottingham grade, *n* (%)			0.584 ¹
I	11 (34.4)	12 (30.0)	
II	19 (59.4)	22 (55.0)	
III	2 (6.3)	6 (15.0)	
Nuclear grade, *n* (%)			0.791 ¹
I	3 (94.0)	5 (12.8)	
II	22 (68.8)	23 (59.0)	
III	7 (21.9)	11 (28.2)	
Molecular subtype, *n* (%)			0.002 ²
Luminal A	24 (75.0)	15 (37.5)	
Luminal B	8 (25.0)	25 (62.5)	
Duration of NET (days), median (p25–p75)	183 (169–212)	189.5 (165–221.5)	0.892 ³
Initial TNM clinical stage group, *n* (%)			0.001 ¹
IA	13 (40.6)	3 (7.5)	
IIA	13 (40.6)	14 (35.0)	
IIB	1 (3.1)	7 (17.5)	
IIIA	2 (6.3)	3 (7.5)	
IIIB	3 (9.4)	13 (32.5)	
ME3 (new 2-tier), *n* (%)			<0.001 ²
<20	29 (90.6)	17 (42.5)	
≥20	3 (9.4)	23 (57.5)	
ME3 (original 3-tier), *n* (%)			0.013 ¹
<18	22 (68.8)	14 (35.0)	
18–31	9 (28.1)	23 (57.5)	
>31	1 (3.1)	3 (7.5)	
ME3 (continuous), median (p25–p75)	16.29 (11.71–18.86)	20.39 (14.60–23.88)	0.001 ³

¹ Fisher’s exact test. ² Chi-square test. ³ Mann–Whitney U test. # PCR/MRD = pathological complete response plus minimal residual disease. ME3 = Magee equation 3 NET = neoadjuvant endocrine therapy.

**Table 3 cancers-16-00339-t003:** Variables related to pathological response in breast cancer patients undergoing NET.

Variables	OR (95% CI)	*p*-Value
Initial clinical stage group		
IA	1.000 (ref.)	Ref.
IIA	0.247 (0.050–1.223)	0.087
IIB	0.042 (0.03–0.562)	0.017
IIIA	0.208 (0.018–2.362)	0.205
IIIB	0.097 (0.014–0.676)	0.018
ME3 (new 2-tier)		
≥20	1.000 (ref.)	Ref.
<20	9.585 (2.291–40.100)	0.002
Post	72	96.0

Legend: ME3 = Magee equation 3; OR = Odds ratio; CI = Confidence interval.

## Data Availability

The original contributions presented in the study are included in the manuscript and Appendix A. Further enquires can be directed to the corresponding author.

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
