# Peer review of "The Magee 3 Equation Predicts Favorable Pathologic Response to Neoadjuvant Endocrine Therapy in Breast Cancer Patients"

_cancers, 2024, doi:10.3390/cancers16020339_

Round 1

Reviewer 1 Report

Comments and Suggestions for Authors

The Magee 3 Equation (ME3) predicts favorable pathologic response to neoadjuvant endocrine therapy (NET)  in breast cancer (BC) patients.

This retrospective study included BC who underwent NET followed surgery. 75 women were eligible to this study, 41 as luminal A.  A new 2-Tier ME3 was significantly pathological response. The authors concluded that ME3 is a cost-effective alternative to Oncotype DX.

MAJOR COMMENTS:

ME3 is an interesting tool and this study could be improved to make easier to read and understanding:

1.      The selection of patients could be reduced : 172 patients but only 75 with a ME3 (to be in agreement with the title)

2.      ME3 and oncotype DX were indicated for BC expressed ER and without lymph node metastasis (N0): unfortunately, the LN status was not expressed (only line 105 “distant metastases that suggest hematogenous metastasis)

3.      In results (table 1 and 2), because a too small date, are not informative and so discarded or to expressed in the text (3 peri/pre vs. 72 post menopausal status), histologic type (65 ductal vs. 10 other) – molecular subtype is preferred-, estrogen receptor (1 negative that could be discarded by definition?!), HER2 (only 2 positive likely luminal B and could be discarded by definition). The small data for ME3 original could be also explain that the authors created the new 2-tier ME3? To explain

4.      We are amazed at the absence of IB or IC clinical stage (TNM) tumors that could be create a bias to the results? The authors could be discussed about this.

MINOR COMMENTS:

1.      It could be interesting to calculate Recurrence free survival curve.

2.      Ductal BC is now called NST invasive carcinoma.

3.      Cut-off of positive ER could be expressed

4.      Table 1 : confusion % and median

5.      Table 1 “suptype” -> “subtype”

6.      Line 268 “ypIA” -> “ypTIa”

Author Response

Reviewer 1

The Magee 3 Equation (ME3) predicts favorable pathologic response to neoadjuvant endocrine therapy (NET)  in breast cancer (BC) patients.

This retrospective study included BC who underwent NET followed surgery. 75 women were eligible to this study, 41 as luminal A.  A new 2-Tier ME3 was significantly pathological response. The authors concluded that ME3 is a cost-effective alternative to Oncotype DX.

MAJOR COMMENTS:

ME3 is an interesting tool and this study could be improved to make easier to read and understanding:

  1. The selection of patients could be reduced : 172 patients but only 75 with a ME3 (to be in agreement with the title)

Response: Thank you for the suggestion. We agree that the way we initially phrased it was poor, making it seem like the selection was too restrictive. To make it clearer to readers, we have modified the sentence, starting from the fact that the potentially eligible cases were considered. In fact, what made it more challenging was the lack of histological grading in Nottingham scores (rather than three categories) and also the assessment of estrogen and progesterone receptors as positive or negative, rather than including intensity and frequency for H-Score calculation. For this project, our goal was to use existing retrospective data, without necessarily reviewing the pathology of all cases      

Previous text: “A total of 172 patients with a history of receiving NET treatment were identified; however, 4 did not undergo surgery, and 52 did not have Ki67 assessment in the initial biopsy. Out of the potential 120 cases, 75 could be classified using ME3, and 61 using other MEs (ME1, ME2, and MEmean), and were included in the study.”

New text: “Out of the potential 120 cases that had received NET and underwent a Ki67 assessment in the initial biopsy, 75 could be classified using ME3, while 61 could be classified using other MEs (ME1, ME2, and MEmean) and were included in the study.”

  1. ME3 and oncotype DX were indicated for BC expressed ER and without lymph node metastasis (N0): unfortunately, the LN status was not expressed (only line 105 “distant metastases that suggest hematogenous metastasis)

Response: Indeed, regarding adjuvant treatment, Oncotype DX has primarily been indicated for tumors with positive hormonal receptors and negative lymph nodes (TAILORX trial). However, in the case of neoadjuvant treatment, Oncotype DX can be used, albeit with less scientific evidence, for selecting cases for neoadjuvant treatment, both with positive and negative lymph nodes. Nonetheless, we agree with the reviewer and have included data related to lymph node status in the paper (please see below).

Previous text: "... The most frequent initial clinical staging was IIA (n=28, 37.3%). Of the total, 41 (54.7%) and 34 (45.3%) were classified as luminal A and luminal B, respectively."

New text: "... The most frequent initial clinical staging group was IIA (n=28, 37.3%), with 26 (34.7%) of the patients having clinically involved regional lymph nodes; 21 (28%) and 5 (6.7%) classified as cN1 and cN2, respectively"

  1. In results (table 1 and 2), because a too small date, are not informative and so discarded or to expressed in the text (3 peri/pre vs. 72 post menopausal status), histologic type (65 ductal vs. 10 other) – molecular subtype is preferred-, estrogen receptor (1 negative that could be discarded by definition?!), HER2 (only 2 positive likely luminal B and could be discarded by definition). The small data for ME3 original could be also explain that the authors created the new 2-tier ME3? To explain

Response: We agree with the reviewer that some of the data provide limited information due to a small sample size. A section of the paragraph describing the case series was modified, moving the more important molecular profile data to an earlier part, and eliminating less informative details that were already included in the table.

In Results, 2nd paragraph:

Previous text:” Out of the 75 patients included in this analysis, all were female (n=75, 100%), with a mean age (standard deviation) of 69.4 (9.98) years. The majority were postmenopausal (n=72, 96%) and had an ECOG-PS of 0/1 (n=66, 88%). Most had a histological subtype of ductal carcinoma (n=65, 86.7%) and Nottingham grade 2 (n=43, 57.3%). The most frequent initial clinical staging was IIA (n=28, 37.3%). Of the total, 41 (54.7%) and 34 (45.3%) were classified as luminal A and luminal B, respectively. Table 1 provides details of the study participants' characteristics.”

New text: ”Out of the 75 patients included in this analysis, all were female (n=75, 100%), with a mean age (standard deviation) of 69.4 (9.98) years. The majority were postmenopausal (n=72, 96%) and had an ECOG-PS of 0/1 (n=66, 88%). Of the total, 41 (54.7%) and 34 (45.3%) were classified as luminal A and luminal B, respectively. The most frequent initial clinical staging group was IIA (n=28, 37.3%), with 26 (34.7%) of the patients having clinically involved regional lymph nodes; 21 (28%) and 5 (6.7%) classified as cN1 and cN2, respectively. Table 1 provides details of the study participants' characteristics”.

Regarding the estrogen receptor, the negative case tested positive for progesterone receptor and was therefore included in the analysis. An additional explanation has been included in the "Methods," section 2.3, "Participants’ accrual and data collection," to clarify the study's eligibility criteria. As for ME3, the traditional cut-offs were of limited utility in this study. This may be partly due to the reduced number of cases with scores above 31. Therefore, we chose to determine the cut-off based on our dataset. The threshold of 20 (close to the original 18) makes clinical and biological sense and could have clinical utility. Further studies are needed to confirm or refute our finding.

Previous text: “Patients aged 18 and older of both genders with invasive carcinoma of any histological subtype who received any NET and subsequently underwent potentially curative surgery were included. Cases with distant metastases, previous invasive breast carcinoma, those operated on in another facility, or those lacking complete information regarding estrogen receptor, progesterone receptor, histological grade, or Ki-67 evaluation were excluded.”

New text: “Patients aged 18 and older of both genders with invasive carcinoma of any histological subtype, showing positive expression of estrogen receptor, and/or progesterone receptor, regardless of HER-2 expression, who received any NET and subsequently underwent potentially curative surgery were included. Cases with distant metastases, previous invasive breast carcinoma, those operated on in another facility, or those lacking complete information regarding estrogen receptor, progesterone receptor, histological grade, or Ki-67 evaluation were excluded.”

  1. We are amazed at the absence of IB or IC clinical stage (TNM) tumors that could be create a bias to the results? The authors could be discussed about this.

Response: Thank you very much for your comment, which helped us understand that the text was not adequately clear. The staging we referred to is the grouped TNM staging, not the T (Tumor) staging alone. Therefore, stage IB would encompass N1mic, which is not common in clinical staging. The grouped clinical stages include stages IA, IB, IIA, IIB, IIIA, IIIB, and IIIC, as well as 0 and IV.

In Results section, lines 183-184:

Previous text: “The most frequent initial clinical staging was IIA (n=28, 37.3%),…”

New text: “The most frequent initial clinical staging group was IIA (n=28, 37.3%),…”

In Table 1, under "Clinical characteristics of patients included in the study," the line "Initial TNM clinical Stage" was modified to "Initial TNM Clinical Stage Group."

In Table 2, the line "Clinical Stage, n%" was changed to "Initial TNM Clinical Stage Group, n%."

Next, in Table 3, the line "Clinical Stage" was changed to "Initial Clinical Stage Group."

MINOR COMMENTS:

  1. It could be interesting to calculate Recurrence free survival curve.

Response: We agree with the reviewer that analyzing recurrence-free survival could provide valuable insights for interpreting the study. However, we currently lack reliable and robust follow-up data. We plan to conduct such an analysis at a later time.

  1. Ductal BC is now called NST invasive carcinoma. 

Response: It was corrected accordingly. In Methods, “2.4 Analysed variables”:

Previous text: “Tumor pathology-related variables: histological type (ductal versus lobular versus mucinous versus other types);…”

New text: “Tumor pathology-related variables: histological type (no special type [NST] versus lobular versus mucinous versus other types);…”

Table 1, line “histologic type”, the term “ductal” was modified to “Ductal / No special type”.

We chose to keep 'ductal' in the table, alongside 'No special type,' separated by a slash, because the term 'ductal' is more widely recognized in the general oncological field, despite being theoretically less appropriate. Furthermore, this is the form described in general non-pathology-specific texts, such as in the NCCN.org Breast Cancer chapter (NCCN Guidelines Version 4.2023).

  1. Cut-off of positive ER could be expressed.

Response: The cut-offs of positive ER and PR were added as requested.

In Methods, “2.4 Analysed variables”:

Previous text:” Tumor pathology-related variables: histological type (ductal versus lobular versus mucinous versus other types); Nottingham histological grade (ranging from 0 to 9 and classified as grades 1, 2, or 3); estrogen receptor (positive versus negative); progesterone receptor (positive versus negative);…”

New text: “Tumor pathology-related variables: histological type (no special type [NST] versus lobular versus mucinous versus other types); Nottingham histological grade (ranging from 0 to 9 and classified as grades 1, 2, or 3); estrogen receptor (positive, if at least 1% of tested cells are positive versus negative); progesterone receptor (positive, if at least 1% of tested cells are positive versus negative);…”

  1. Table 1 : confusion % and median

Response: We appreciate the reviewer's comment. We have chosen to remove the age information from the table and keep it with complete data only in the text. Similarly, we have modified the table to make the interpretation of Magee's equation scores more obvious.

  1. Table 1 “suptype” -> “subtype”

Response: It was corrected

  1. Line 268 “ypIA” -> “ypTIa”

Response: Thank you for the comment. We have modified previous texts (please see response to comment #2) to clarify that it refers to the grouped TNM stage.

We sincerely hope that we have adequately addressed any questions or concerns.

Reviewer 2 Report

Comments and Suggestions for Authors

The authors report an interesting evaluation of the Magee equations as potential predictive factors of response to neoadjuvant endocrine therapy in early HR+ breast cancer.

the paper is well written and carries a clear message. Some remarks along the text :

l 141 sqq : please define in the Methods section what are the "new tiers" of ME3, and how these tiers refer to a potential RS value

l 149 : misspelling "CRP/MRD"

l 232 in the table : "post/72/96.0" should be removed (or explained !)

I also have some general important remarks

- the endpoint PCR/MRD is not a generally accepted endpoint, and this shoud be further discussed. In the field of neoadjuvant endocrine therapy, alternative endpoints are also considered, such clinical/US/MRI response, and biological response sucha as Ki67 decrease and the PEPI Scores. The limitations have been clearly acknowledged regarding the biological response, they must be adressed also with regard to the clinical response

- as suggested, please comment the relationship in the discussion between ME3 and RS

- the discussion should also refer to the value of the Magee equations as predictors of response to neoadjuvant chemotherapy (please see : PMID: 28548119 and 32661297)

Author Response

Reviewer 2

The authors report an interesting evaluation of the Magee equations as potential predictive factors of response to neoadjuvant endocrine therapy in early HR+ breast cancer.

the paper is well written and carries a clear message. Some remarks along the text :

l 141 sqq : please define in the Methods section what are the "new tiers" of ME3, and how these tiers refer to a potential RS value

Response: Thank you for the valuable comment. Indeed, there was a need for a clearer explanation in the Method section.

In Methods Section, “2.4 Analysed variables”:

New text: “The MEs were analyzed in three different ways: as a continuous variable, based on three categories from previous studies (original 3-tier; <18 vs. 18-31 vs. >31), and using a cutoff defined by the present analysis into two categories (new 2-tier, < or > the best cutoff point determined by the ROC curve).”

l 149 : misspelling "CRP/MRD"

Response: Thanks! It was corrected accordingly.

l 232 in the table : "post/72/96.0" should be removed (or explained !)

Response: The Table 1 has been edited for enhanced clarity. The entry 'post/72/96.0' indicates the count and percentage of postmenopausal patients in the study sample (72 out of 75 patients). As expected and confirmed by a median age of 69 years, the majority of patients treated with NET were in the postmenopausal stage.

I also have some general important remarks

- the endpoint PCR/MRD is not a generally accepted endpoint, and this shoud be further discussed. In the field of neoadjuvant endocrine therapy, alternative endpoints are also considered, such clinical/US/MRI response, and biological response sucha as Ki67 decrease and the PEPI Scores. The limitations have been clearly acknowledged regarding the biological response, they must be adressed also with regard to the clinical response.

Response: We appreciate the highly relevant comment. Indeed, other methods for assessing response exist, especially with the use of the PEPI score. Additionally, clinical response has also been widely used in research and clinical practice. Given that this is a retrospective analysis, we opted for a more robust and reliable endpoint, as clinical response assessment in the analyzed cases was not conducted in a standardized manner, as would be the case in a prospective cohort study or clinical trial. Many cases had their response defined only by means of caliper measurements, while others relied on ultrasound, MRI, mammograms, or CT scans.

The following phrases were added in the Discussion, Limitations:

Added text: “Furthermore, response assessments can be carried out through physical examination (using calipers), breast ultrasound, and breast MRI. However, such response evaluation may be biased in retrospective studies where the chosen examination and timing of assessment were not planned in advance.”

- as suggested, please comment the relationship in the discussion between ME3 and RS

- the discussion should also refer to the value of the Magee equations as predictors of response to neoadjuvant chemotherapy (please see : PMID: 28548119 and 32661297)

Response: We are grateful for the critics, as we believe the text did indeed become more comprehensive following the reviewer's suggestions. The following paragraph was added in the Discussion section (4th paragraph).

New paragraph: “A previous study using pre-treatment tumor biopsies have demonstrated a positive correlation (Rho = 0.58) between the continuous scores of MEs and ODX-RS (Soran et al., 2020). In a sample of low-grade invasive BCs, commonly hormone-sensitive, the overall agreement between ODX-RS and MEs risk categories was 68.7% (Hou et al., 2017). While not yet assessed in the NET setting, previous studies have confirmed the predictive role of MEs (particularly ME3) in the neoadjuvant chemotherapy scenario. Farrugia et al. (2017) identified that patients with ME3 >31 were 13 times more likely to achieve a PCR compared to those with ME3 scores less than 31. A subsequent multicenter study (Bhargava et al., 2021) found that the PCR rate was 0% in ME3 < 18, 0% in ME3 18-25, 14% in ME3 > 25 to <31, and 40% in ME3 scores of 31 or higher. Tumors with a lower likelihood of responding to chemotherapy are precisely those where we observe a higher likelihood of response to NET, enhancing the practical potential of using ME3 in selecting patients for neoadjuvant treatment.

Consequently, four new references were added (two of them suggested by the reviewer). Please see below.

Soran, A.; Tane, K.; Sezgin, E.; Bhargava, R. The Correlation of Magee EquationsTM and Oncotype DX® Recurrence Score From Core Needle Biopsy Tissues in Predicting Response to Neoadjuvant Chemotherapy in ER+ and HER2- Breast Cancer. Eur J Breast Health 2020, 16, 117–123, doi:10.5152/ejbh.2020.5338.

Hou, Y.; Zynger, D.L.; Li, X.; Li, Z. Comparison of Oncotype DX With Modified Magee Equation Recurrence Scores in Low-Grade Invasive Carcinoma of Breast. Am J Clin Pathol 2017, 148, 167–172, doi:10.1093/ajcp/aqx059.

Farrugia, D.J.; Landmann, A.; Zhu, L.; Diego, E.J.; Johnson, R.R.; Bonaventura, M.; Soran, A.; Dabbs, D.J.; Clark, B.Z.; Puhalla, S.L.; et al. Magee Equation 3 Predicts Pathologic Response to Neoadjuvant Systemic Chemotherapy in Estrogen Receptor Positive, HER2 Negative/Equivocal Breast Tumors. Mod Pathol 2017, 30, 1078–1085, doi:10.1038/modpathol.2017.41.

Bhargava, R.; Esposito, N.N.; OʹConnor, S.M.; Li, Z.; Turner, B.M.; Moisini, I.; Ranade, A.; Harris, R.P.; Miller, D. V; Li, X.; et al. Magee EquationsTM and Response to Neoadjuvant Chemotherapy in ER+/HER2-Negative Breast Cancer: A Multi-Institutional Study. Mod Pathol 2021, 34, 77–84, doi:10.1038/s41379-020-0620-2.

Reviewer 3 Report

Comments and Suggestions for Authors

The topic is interesting.  The contribution is clear and the novelty is good.  The paper is well written and orginzed. The manuscript is accepted in the current form. 

Author Response

Reviewer 3

The topic is interesting.  The contribution is clear and the novelty is good.  The paper is well written and orginzed. The manuscript is accepted in the current form. 

Response: Thank you for taking the time to review our manuscript and for your positive evaluation of it.

Reviewer 4 Report

Comments and Suggestions for Authors

Magee equations (MEs) are a set of multivariable models that use clinical and pathologic factors to predict the likelihood of breast cancer recurrence and the need for chemotherapy. In this paper, the authors investigated the potential of MEs as a predictive tool to identify strong responders to neoadjuvant endocrine therapy (NET) in patients with luminal breast cancer. The results showed that ME3, one of the Magee equations, emerged as a promising predictor with an AUC of 0.734, effectively identifying patients likely to respond well to NET.  The use of ME3 as a predictor may be a cost-effective alternative to Oncotype DX.  Overall, the paper provides a promising avenue for improving breast cancer treatment and patient outcomes.

The paper is very well written (just correct a typo in line 49: : isa --> is) and addresses a problem of utmost relevance and timing, so I strongly recommend its publication.

Author Response

Reviewer 4

Magee equations (MEs) are a set of multivariable models that use clinical and pathologic factors to predict the likelihood of breast cancer recurrence and the need for chemotherapy. In this paper, the authors investigated the potential of MEs as a predictive tool to identify strong responders to neoadjuvant endocrine therapy (NET) in patients with luminal breast cancer. The results showed that ME3, one of the Magee equations, emerged as a promising predictor with an AUC of 0.734, effectively identifying patients likely to respond well to NET.  The use of ME3 as a predictor may be a cost-effective alternative to Oncotype DX.  Overall, the paper provides a promising avenue for improving breast cancer treatment and patient outcomes.

Response: Thank you for taking the time to review our manuscript and for your positive evaluation of it.

The paper is very well written (just correct a typo in line 49: : isa --> is) and addresses a problem of utmost relevance and timing, so I strongly recommend its publication.

Response: Thanks! It was corrected.

Round 2

Reviewer 1 Report

Comments and Suggestions for Authors

none comments